computer modelling and simulation/pattern recognition/e-science

cryptocurrency, topic modelling, Latent Dirichlet Allocation, news media, text analysis, sentiment

**Author for correspondence:**
Kelly Ann Coulter
e-mail: k.coulter@ucl.ac.uk

# The impact of news media on Bitcoin prices: modelling data driven discourses in the crypto-economy with natural language processing

## Kelly Ann Coulter

School of Management, University College London, Level 38 and Level 50, One Canada Square, London E14 5AA, UK

(iD) KAC, 0000-0003-3426-0962

This paper examines the relationship between events reported in international news via categorical discourses and Bitcoin price. Natural language processing was adopted in this study to model data-driven discourses in the crypto-economy, specifically the Bitcoin market. Using topic modelling, namely Latent Dirichlet Allocation, a text analysis of cryptocurrency articles ($N = 4218$) published from 60 countries in international news media identified key topics associated with cryptocurrency in the international news media from 2018 to 2020. This study provides empirical evidence that across the corpora of international news articles, 18 key topics were framed around the following categorical macro discourses: crypto-related crime, financial governance, and economy and markets. Analysis shows that the identified discourses may have had a 'social signal' effect on movements in the crypto-financial markets, particularly on Bitcoin's price volatility. Results show these specific discourses proved to have a negative effect on Bitcoin's market price, within 24 h of when the crypto news articles were published. Further, the study found that in some cases, the source of the news may have amplified the volatility effect, particularly in terms of geographical region, relative to broader market conditions.

## 1. Introduction

Cryptocurrency research has primarily focused on Bitcoin, where there has previously been extensive research in analysing the link between Google search volumes and Bitcoin metrics [1–8] and further research developed on the words and sentiment that underlie search terms [9]. While past research has focused on

Google searches and specifically social media, this article shows that alternatively, media such as digital news articles from online news outlets (for example, such as Bloomberg and FT among others), can identify not only popular single worded terms, but contextualized cryptocurrency market sentiment too. By linking words to other common and popular crypto-economic terms (or topics), narratives emerge that express a range of micro ideologies and speculation. The combinations of these narratives cumulate into macro discourses within the crypto-economy, providing a clearer and broader view of overall sentiment.

Bitcoin was the original blockchain. It was the first of a peer to peer digital currency that eliminated the need for a third party (such as a central bank) to validate transactions, by using a distributed ledger system where the decentralized consensus mechanism—Proof of Work—ensured validity and trust within the network [10]. Since Bitcoin's inception, other cryptocurrencies have entered the crypto market. As of February 2021, there were 4501 crypto coins [11]. Cryptocurrencies, or 'crypto-assets' as they are commonly referred to [12], are along with Bitcoin, part of a wider crypto market [13]. It is a market that is growing in the capacity of coins and in trading volume [11] and is an economy with a growing market capital [14].

## 1.1. Current literature

The function of the media is a source of information and sentiment in the financial market [15]. Bloomberg or Reuters, for example, as reputable financial media outlets, can affect the markets as investor behaviour can be in response to company news and events which gain high media coverage [16,17]. Further, social signal effect research of the social media platform Twitter has revealed that increases in opinion polarization and exchange volume precede rising Bitcoin prices, and that emotional valence precedes opinion polarization and rising exchange volumes [6]. Mai *et al.*'s [18] findings conferred that social media sentiment is an important predictor in determining Bitcoin's valuation but highlighted how not all social media messages are of equal impact on Bitcoin's price. For example, the authors showed how social media's effect on Bitcoin are driven primarily by the silent majority, the 95% of users who are less active and whose contributions amount to less than 40% of total messages.

Media coverage that exhibits varying optimism and pessimism may be captured through the fixed effects, as well as article length, writing style and availability of information to different journalists [15]. What discourses the media present to their audiences (its construction), and how the news media presents cryptocurrency (its analysis), are as important and influential to the crypto-economy as Google searches and social media Bitcoin metrics [9]. Research has confirmed the suggestion that movements in financial markets, and movements in financial news are intrinsically interlinked [19]. Sentiment analysis in the media can therefore be particularly useful for computational finance, where digital traces of human behaviour offer great potential to drive trading strategies [6].

Studies have shown that both the informational and affective aspects of news affect the markets in profound ways, impacting on volumes of trades, stock prices, volatility and even future firm earnings [20,21]. Recent research has identified that the interaction between media sentiment and the Bitcoin price exists where there is a correlation between the abnormal returns of Bitcoin and the amount of news media articles published on a daily basis [22]. Karavelicius *et al.* [23] have confirmed the interaction between media sentiment and Bitcoin price, but also have shown that there is a tendency for investors to overreact on news in a short period of time. Under reaction of stock prices to news such as earnings announcements, and over-reaction of stock prices to a series of good or bad news is well documented by Barberis *et al.* [24] as regularities among investor behaviour in how beliefs are formed.

Corbet *et al.* [25] constructed a sentiment index based on news stories that follow the announcements of four macroeconomic indicators: GDP, unemployment, consumer price index (CPI) and durable goods. They found that increases in positive news after unemployment and durable goods announcements result in a decrease in Bitcoin returns. Conversely, they discovered an increase in the percentage of negative news surrounding these announcements is linked with an increase in Bitcoin returns. News relating to GDP and CPI were not found to have any statistically significant relationships with Bitcoin returns. Similarly, Mai *et al.* [26] examined the dynamic relationships between social media and Bitcoin performance. They considered the distinct effects of different social media platforms and different user groups subdivided by posting volume. The results suggested that more bullish forum posts had a positive effect on Bitcoin returns, and the effect was stronger when they only included the posts by users who are less likely to contribute.

Asymmetric volatility in response to news sentiment is not a new research finding. For example, other commodities such as gold (futures) have been found to be more sensitive to negative news causing

volatility [27]. Bitcoin's price sensitivity to material events, it has been found by Feng *et al.* [28], make informed trading very profitable in the crypto market. They summarized that when examining the timing of informed trades, informed traders prefer to build their positions two days before large positive events, and one day before large negative events. The authors concluded that profits of informed trading in the Bitcoin market are estimated to be considerably large.

Google trend popularity along with social media (reddit) analysis has shown the link between words and Bitcoin metrics [1–8]. Google search term frequency has been used as a proxy for attractiveness (or popularity) of crypto to discover potential price drivers. Sovbetov's research [29] in particular observed the attractiveness proxied by Google search term frequency finding significant coefficients for Bitcoin, Ethereum, Litecoin and Monero at 10% significant level. It indicated that 1 unit increase in Google trend popularity of Bitcoin, Ethereum, Litecoin and Monero leads to 1.27, 0.24, 0.07 and 0.05 unit increases in their prices in the long-run, respectively [29].

If Google search terms alone can prove to have this affect, it is likely that use of terms together can provide a narrative effect when words are linked in a contextualized manner. In the following discussion, the natural language processing (NLP) text analysis results are explored thematically in terms of their discourse and their respective market reactions in terms of Bitcoin price. If other commodities such as gold futures can exhibit volatility asymmetrically to news sentiment [27], Bitcoin may also too exhibit such features. This study, however, focuses purely on the negative sentiment and volatility. This research therefore examines the news media and their reporting of cryptocurrency to identify popular discourses and underlying sentiment. By undertaking text analysis of $N = 4218$ cryptocurrency articles published between 2018 and 2020, discourses in the international news media from over 60 countries are modelled using a NLP method, specifically Latent Dirichlet Allocation (LDA) Topic Modelling [30]. LDA is adopted to identify cryptocurrency discourses that may have a social signal effect on movements in the financial markets.

## 1.2. Contributions of this article with Latent Dirichlet Allocation topic modelling

This study provides empirical evidence that key topics associated with the cryptocurrency Bitcoin in the international news media from 2018 to 2020 are framed around the following key macro discourses: crypto-related crime, financial governance, and economy and markets. This research develops Burnie & Yilmaz's [9] research into social media signals and Bitcoin metrics, but instead of determining which words on reddit matter as the Bitcoin pricing dynamic changes from one phase to another, this study takes a complementary next step looking at the broader perspective of overall sentiment and of which are involved in supporting macro discourses in international news media, and the potential effect these discourses have on the Bitcoin price. Where past research has shown a clear connection between the role of the media and the market, i.e. commonly used words and market behaviour [15–17,19], discourses where words present context, may have differing effects. This research identifies those key crypto discourses and maps them to the Bitcoin market in a novel, and yet unexplored way in the crypto-economy.

LDA topic modelling is one technique in the field of NLP text mining that is a process to automatically identify topics present in a text object and to derive hidden patterns exhibited by a text corpus, based on probabilistic latent semantic analysis [31]. Using Bayesian inference, a topic therefore, is a distribution over a fixed vocabulary, where unobserved (or latent) topics are assumed to be generated first before documents. Documents are generated from a mix of topics in different proportions, and in this way, it is understood as a generative process [32]. A key assumption of LDA topic modelling is that documents can exhibit multiple topics but only the number of topics is specified in advance, it is hence a generative process.

Blei [32] argued for the use of topics modelling, claiming that the 'utility of topic models stems from the property that the inferred hidden structure resembles the thematic structure of the collection'. Adopting probabilistic topic models for text analysis as an algorithmic solution can be therefore very useful for the purpose of document clustering, organizing large blocks of textual data, information retrieval from unstructured text and feature selection [33]. This study adopted computational text analysis and human qualitative interpretation for the discourse analysis. In the first instance, the quantitative work was performed using the Python programming language using NLP to undertake the unsupervised learning technique of LDA [34] topic modelling. No topics were given or 'fixed' to the model, as would have been the case in an alternative supervised approach.

The unsupervised approach was used in this crypto study for finding and observing co-occurring groups of words, understood as thematically coherent 'topics' in large clusters of texts. Topics can be defined as a repeating pattern of co-occurring terms in a corpus; however, the name can be

misleading. As Jacobs & Tschötschel [35, p. 471] explained: 'topics are clusters of words that reappear across texts, but the interpretation of these clusters as themes, frames, issues, or other latent concepts (such as discourses) depends on the methodological and theoretical choices made by the analyst'. Iterative qualitative interpretation was then required from the researcher to undertake a discourse analysis, which presented as the most frequently co-occurring across the corpus, understood as the key topics. Both computationally and qualitatively, the data were pre-processed before analysis, which broadly involved cleaning and text processing the data.

# 2. Data preparation

## 2.1. Data collection and preparation

To construct a suitable corpus of documents for analysis, the researcher manually collected and downloaded media articles in the form of text files from traditional media outlets. The articles were retrieved from across 60 countries globally, covering the broad theme of 'cryptocurrency'. Four thousand two hundred eighteen news articles written in the English language were drawn from the Nexis news database and 'News API' [36], using the query 'cryptocurrency'.

## 2.2. Natural language pre-processing stage

After the text had been collected and collated, the text was pre-processed in Python using the SpaCy [37], Gensim [38] and Pandas [39] python packages. Pre-processing was a prerequisite prior to conducting the NLP on the text. The NLP stage essentially consisted of four broad steps: (1) to load the input data (crypto text articles), (2) to pre-process the data, (3) to transform documents into bag-of-words vectors and finally (4) to train the LDA model.

SpaCy is a free, open-source library for advanced NLP in Python. SpaCy explains that it is '… designed specifically for production use and helps you build applications that process and 'understand' large volumes of text. It can be used to build information extraction or natural language understanding systems, or to pre-process text for deep learning' [37]. SpaCy was used to 'parse and tag' a given document. This was where the trained pipeline and its statistical models were applied, enabling SpaCy to make predictions of which tag or label was most likely to apply in the context. One of SpaCy trained components included binary data that was produced by showing the corpus enough examples for it to make predictions that generalized across the language—for example, a word following 'the' in English was most likely to be a noun.

Part of the pre-processing stage was to train the phraser which automatically detected common phrases (multi-word expressions) from a stream of sentences. This process included lemmatizing the text articles (assigning the base forms of words [40]) using SpaCy, tokenizing the text articles (segmenting the text into words and punctuation marks etc. [41]) and to compute bigrams (multi-word expressions or common phrases) using Gensim [38].

# 3. Methodology

## 3.1. Step 1: NLP stage

Gensim was used to vectorize the sets of tokens into a doc-term matrix which were then used to determine the LDA model. Therefore, after the text was converted from text to tokenized documents, the 'tokens' were stored in a dictionary format to create a map between words and their integer ID's, in an ID-to-word fashion. This then allowed for the tokenized documents to be converted to vectors. An algorithm (doc2bow) was then applied to count the number of occurrences of each distinct word, converting the word to its integer and returning the result as a sparse vector. In the first trial, the LDA model was then set up and run over an arbitrary 20 topics in the first instance.

## 3.2. Step 1: determining the number of topics in the LDA model

An assumption about LDA is that the number of topics is assumed known and fixed [32]. The Bayesian non-parametric topic model [42] provides a solution where the number of topics is determined by the collection during posterior inference, and new documents can exhibit previously unseen topics [32].

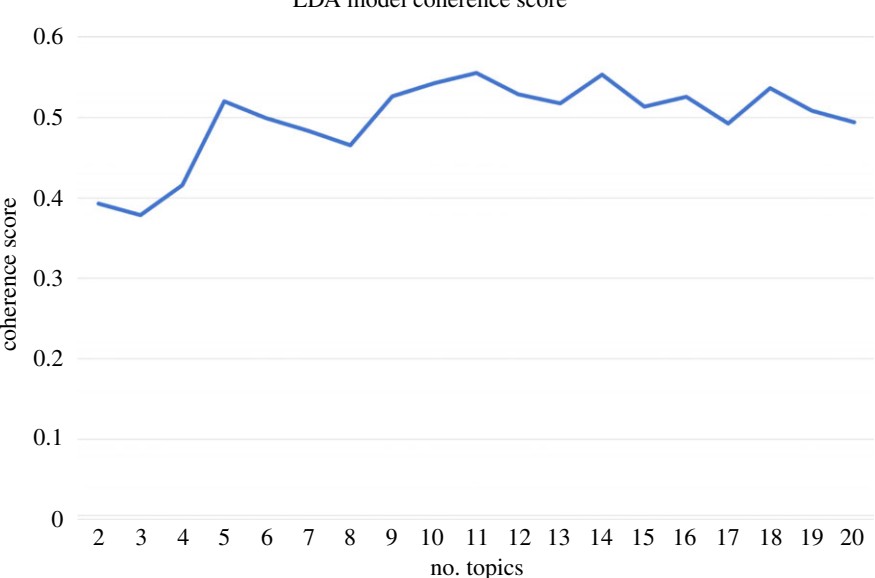

**Figure 1.** LDA model coherence scores.

Therefore, the Gensim implementation of LDA was adopted. Topic coherence was used as an intrinsic evaluation metric in this study. This metric was used to quantitatively justify the model selection. Topic coherence measures score a single topic by measuring the degree of semantic similarity between high scoring words in the topic. These measurements help distinguish between topics that are semantically interpretable topics and topics that are artefacts of statistical inference [43].

Human inference is superior for topic interpretability, as 'human topic rankings serve as the gold standard for coherence evaluation' [44]. However, human evaluations can be a lengthy and time-consuming process. Therefore, quantitative topic coherence measures aided in the process for this big data study. As there is no 'correct' number of topics per se in topic modelling, some results may identify better topics than others. In this study, it was then a reiterative process to conduct various trials of number of topics to identify the optimal number. To find the optimal number of topics, LDA trials based on different values of number of topics were undertaken, to select the one that produced the highest coherence score alongside initiative assessment.

## 3.3. Step 2: coherence scores

A graph of coherence scores is presented in figure 1. It was qualitatively apparent from topic model results that the optimal number of topics in the LDA model trials was 18 topics; this was the third highest coherence score (figure 1). The top three coherence scores were extremely close in value and a decision was taken by the researcher to commence with the 18-topic model. The LDA model was then run, passing in the default alpha and beta values to calculate coherence.

## 3.4. Step 3: sampling stage for price change evaluation

Bitcoin data was sourced from messari.io to establish percentage price change in 24 h. Daily returns were calculated over a two-year period. Using Excel, the data were ranked from highest percentage change to lowest percentage change. The lowest 5% of these observations constitute the subsample, which is presented in the below charts.

## 3.5. Step 4: categorization of the media articles by discourse

To label the media articles according to their thematic category (or discourse type), the researcher qualitatively assessed the 18-topic model to consider the broad themes that the model had discovered. Following this exercise, the researcher manually recorded each news article in the price change subsample according to its content theme with reference to the themes produced in the topic model. The news media articles from the Bitcoin price subsample were subsequently categorized into three broad

but distinct generalized categories that featured in the topic model: crypto-related crime discourse (labelled crypto-crime), financial governance and regulation discourse (labelled crypto-governance), and finally economic discourse (labelled economy and market).

# 4. Results

## 4.1. 18-topic LDA model and categorizing crypto news articles

In a two-step process, this topic modelling algorithm estimated the distribution of topics over a set of documents and a probability distribution of words for each of the 18 topics shown in figure 2. Therefore, the number next to each topic represents the topic-word probability distribution across the corpus. For every word then there is a proportion expressed as a score aligned to each topic. This is a distinguishing characteristic of LDA; the documents in the selection share the same set of topics, but each document exhibits those topics in different proportion [32]. The 18-topic model was run using the default alpha and beta scores. The model produced top topics, as presented in figure 2. The top topics indicated that there were three predominant discourses present in the dataset—cryptocurrency discourses related to (1) crime, (2) financial governance and (3) economic market sentiment. These established the themes into which the subsample of news articles would be categorized, so that news story discourse could be assessed against on potential effect on Bitcoin market price.

## 4.2. Crypto media article's published and associated highest percentage negative price changes in Bitcoin 2018 and 2019

The charts displayed below show the 5% most negative changes in a two-year period. Each data point represents a news media article published, which is categorized by discourse and is labelled with publication date and Bitcoin price change.

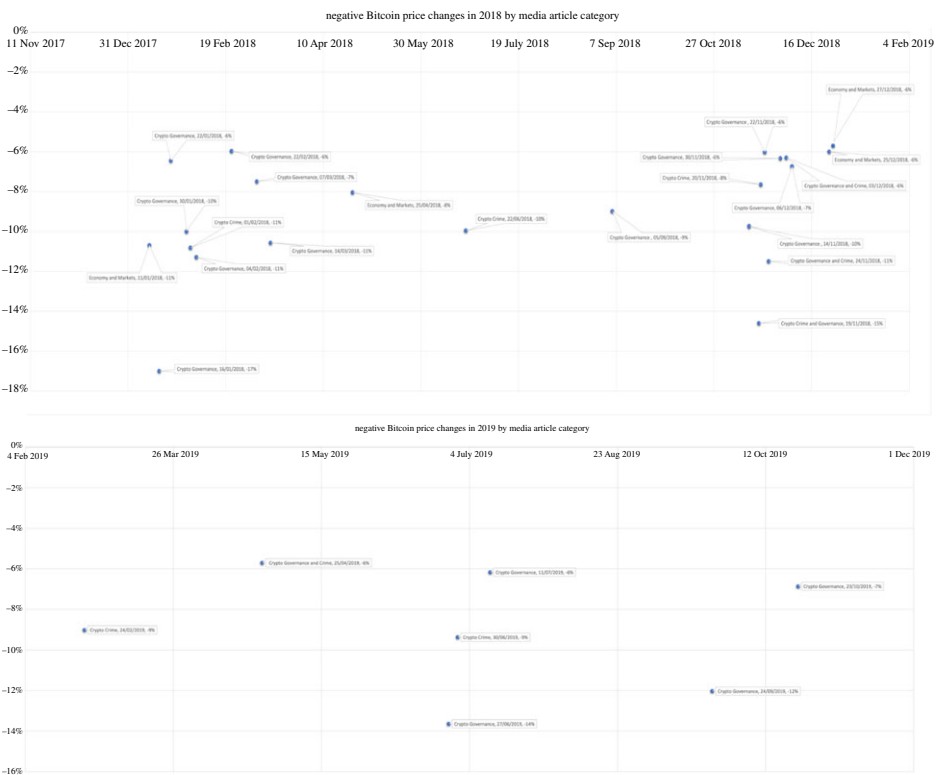

TOPIC 0 | 0.027*"ph" + 0.003*"vh" + 0.002*"group" + 0.002*"victor_harbor" + 0.002*"club" + 0.002*"centre" + 0.002*"carrickalinga_house" + 0.002*"goolwa" + 0.002*"market" + 0.001*"pt_elliot"

TOPIC 1 | 0.002*"quadrigacx" + 0.001*"court_appoint" + 0.001*"payment_processor" + 0.001*"bank_draft" + 0.001*"vancouver_base" + 0.001*"scotia_supreme" + 0.001*"owe" + 0.001*"ceo_and_sole" + 0.001*"pass_code" + 0.001*"quadrigacx_user"

TOPIC 2 | 0.006*"year" + 0.005*"people" + 0.005*"company" + 0.005*"use" + 0.004*"work" + 0.004*"time" + 0.004*"money" + 0.003*"know" + 0.003*"day" + 0.003*"like"

TOPIC 3 | 0.000*"patent" + 0.000*"commend" + 0.000*"philips" + 0.000*"hku_space" + 0.000*"asia_leadership" + 0.000*"legal_team" + 0.000*"wuxi" + 0.000*"chong_sing" + 0.000*"jeepney" + 0.000*"kokila_alagh"

TOPIC 4 | 0.001*"maren" + 0.000*"axe" + 0.000*"ueland" + 0.000*"bobby" + 0.000*"khayali" + 0.000*"esalen" + 0.000*"rodney" + 0.000*"ejjoud" + 0.000*"jespersen" + 0.000*"ueland_and_jespersen"

TOPIC 5 | 0.001*"ceza" + 0.001*"iceland" + 0.001*"lambino" + 0.000*"salerno" + 0.000*"char" + 0.000*"economic_zone" + 0.000*"cagayan_economic" + 0.000*"zone_authority" + 0.000*"raul_lambino" + 0.000*"ihe"

TOPIC 6 | 0.016*"bitcoin" + 0.008*"cryptocurrency" + 0.008*"blockchain" + 0.007*"use" + 0.006*"cryptocurrencie" + 0.006*"company" + 0.006*"year" + 0.006*"market" + 0.006*"new" + 0.005*"technology"

TOPIC 7 | 0.016*"wright" + 0.005*"nakamoto" + 0.004*"north_korea" + 0.002*"north_korean" + 0.002*"craig_wright" + 0.001*"satoshi" + 0.001*"mr_freeman" + 0.001*"andresen" + 0.001*"wright_claim" + 0.001*"pty_ltd"

TOPIC 8 | 0.004*"hagen" + 0.001*"anne_elisabeth" + 0.001*"tom_hagen" + 0.001*"ransom_note" + 0.001*"disappearance" + 0.000*"falkevik_hagen" + 0.000*"mr_hagen" + 0.000*"oslo" + 0.000*"hagen_lawyer" + 0.000*"char"

TOPIC 9 | 0.000*"char" + 0.000*"bitcoin_btc" + 0.000*"let_have_a_baby" + 0.000*"million_yuan" + 0.000*"global_stablecoin" + 0.000*"eur_million" + 0.000*"facebooks" + 0.000*"week_edition" + 0.000*"today_where_satoshi" + 0.000*"satoshi_nakaboto"

TOPIC 10 | 0.014*"good_morning" + 0.010*"property" + 0.009*"euro" + 0.008*"income" + 0.008*"thank" + 0.008*"box" + 0.007*"rent" + 0.006*"greeting_and_a_lot" + 0.006*"declare" + 0.005*"return"

TOPIC 11 | 0.001*"oil_and_gas" + 0.001*"intercontinental_exchange" + 0.000*"loeffler" + 0.000*"char" + 0.000*"energy_sector" + 0.000*"sugarbud" + 0.000*"security_filing" + 0.000*"kolochuk" + 0.000*"kelly_loeffler" + 0.000*"corporate_governance"

TOPIC 12 | 0.000*"char" + 0.000*"shop_locally" + 0.000*"teach_young" + 0.000*"stitcher" + 0.000*"second_be_decentralisation" + 0.000*"retailer_which_own_no_inventory" + 0.000*"undermine_by_communication" + 0.000*"chairman_of_wandisco" + 0.000*"unassailable_have_be_superseded" + 0.000*"bygone_age"

TOPIC 13 | 0.000*"â" + 0.000*"manifesto" + 0.000*"tarrant" + 0.000*"char" + 0.000*"mass_murderer" + 0.000*"australian_academic" + 0.000*"pseudocommando" + 0.000*"ammunition_belt" + 0.000*"tarrant_life" + 0.000*"regular_guy"

TOPIC 14 | 0.003*"accuse" + 0.001*"crore" + 0.001*"arrest" + 0.001*"r_crore" + 0.001*"kotadiya" + 0.001*"bhatt" + 0.001*"surat" + 0.001*"patel" + 0.001*"police" + 0.001*"bhardwaj"

TOPIC 15 | 0.000*"char" + 0.000*"gramatik" + 0.000*"epigram" + 0.000*"data_source" + 0.000*"muvhango" + 0.000*"napier" + 0.000*"bitcoin_btc" + 0.000*"ada" + 0.000*"skeem_saam" + 0.000*"isidingo"

TOPIC 16 | 0.000*"gv" + 0.000*"sept" + 0.000*"oct" + 0.000*"char" + 0.000*"nando" + 0.000*"camping" + 0.000*"macaron" + 0.000*"hulme" + 0.000*"info" + 0.000*"admission"

TOPIC 17 | 0.009*"venezuela" + 0.007*"petro" + 0.004*"maduro" + 0.003*"bolivar" + 0.003*"oil" + 0.003*"venezuelan" + 0.002*"venezuelans" + 0.002*"sovereign_bolivar" + 0.002*"economic" + 0.002*"hyperinflation"

**Figure 2.** LDA topics.

# 5. Discussion

## 5.1. The relationship between discourse and Bitcoin price

The first observation from the sample of the most negative Bitcoin market changes is that cryptocurrency news media articles pre-empted the price change within 24 h prior. The news articles published contained specifically the following generalized discourses: crypto-related crime, crypto-related governance or crypto-economic speculation. This holds true across the two-year period sampled. These charts show the 5% most negative price changes in a two-year period, supporting the

presupposition that these generalized discourses are sensitive themes, which have great effects on Bitcoin's volatility. These discourses are particularly associated with a negative market impact and are further discussed below, each discourse in turn.

## 5.2. The crypto-crime discourse

The LDA output identified topics 1, 8, 10 and 14 specifically as relating to a crypto-crime discourse important in news media (figure 2, LDA topics). Broadly across the entirety of the news corpora in the dataset, the crypto-crime discourse can be described as a generalized category which contains news media articles which refer to cryptocurrency-related crime. These are for example news coverage on crypto scams, scandals or crypto exchange hacks. Publicity through news media reports comprising discussion of these events encompassing a criminal element appear to have a negative price effect on Bitcoin (see above graphs and further descriptions below).

Particularly of note, for example, were news stories reporting on QuadrigaCX, Canada's largest former cryptocurrency exchange. In 2019, the exchange ceased operations and the company was declared bankrupt with C\$215.7 million in liabilities and about C\$28 million in assets, with the FBI and Royal Canadian Mounted Police investigating due to the mysterious death of Quadriga's CEO, Gerald Cotton [35]. There were numerous media reports on this case in this study's news corpora which led it to become an important topic within the crypto-crime discourse. This is demonstrated in the LDA output as topic 1. Specifically, however, a news media article released on 24 February 2019 reported the development of this story, where the news was released that a judge issued an order for disbursement of \$30 million in bank drafts tied to the insolvency case. The publication of this news article coincided with a −9% price change in the Bitcoin market.

The Quadriga crypto scandal, along with other generalized crypto-crime events, clearly can play into narratives of crypto-crime which may cause negative market confidence. After each group of articles were published in these specific instances in the Canadian press, the price of Bitcoin dropped each time in varying percentages. While there are many factors that can cause price fluctuations in financial markets, the function of the media is a consistent source of information and sentiment [15]. Sentiment therefore remains a candidate for causes of crypto volatility, due to investor behaviour reacting to negative news gaining high media coverage [16,17]. The weight of the negative sentiment in the Canadian press may reflect the fact that North America is the third-most active region by cryptocurrency volume moved on-chain, just behind Northern & Western Europe [45]. Therefore, news concentrated in this area may target large audiences in the crypto sector, which in turn affects a larger amount of investor behaviour in these regions.

This led the researcher to question how other important topical events related to cryptocurrency crime reported in international news media influenced Bitcoin price. Other crypto-crime events that coincided with Bitcoin market price drops were the event of the theft from Coincheck. Coincheck was one of Japan's largest digital currency exchanges and as a consequence of the hacking attack on its network, reportedly lost \$534 m (£380 m) worth of virtual assets [46]. On 1 February 2018, a media outlet reported that the aggregate market value of cryptocurrencies dropped by roughly 10% upon the news of the massive theft at Coincheck. On the actual day of the publication of the news article covering the theft, the graph above actually shows a −11% drop in Bitcoin value. Similarly, in another crypto exchange attack on Seoul-based Bithumb, news media reported that 35bn won (£24 m; \$31.6 m) worth of cyber-cash had been 'seized' overnight [47]. On 22 June 2018, news media reported this attack, which coincided with a −10% Bitcoin price change. Other media news of crypto exchange hacks that appear to have had a negative impact on Bitcoin's price include coverage of the Cryptopia hack and the subsequent liquidation court case. A news article published on 10 April 2020 that covered the Cryptopia court case coincided with a −6% Bitcoin price drop.

Further on the topic of crypto-crime reporting, on 30 June 2019 the Financial Times reported a warning on crypto scams. This story simultaneously occurred alongside another crypto-crime news reports of the Iranian Authorities shutting down crypto mining farms. On this day, Bitcoin lost −9% of its value. Similarly, a Bitcoin scam that was reported in the international press on 10 April 2020 corresponded with −6% Bitcoin price fall. From the examples provided with evidence displayed on the above graphs, findings indicate that crypto-crime events[1] which are reported in international press have a negative impact on the Bitcoin market. This may be due to widespread concern about the security and the trustworthiness of Bitcoin and may cause its legitimacy to be brought into question when it is linked to criminal activity reported in the media. Concerns about Bitcoin's reputation as a reliable payment tool or

---

[1]Those events that connect or associate cryptocurrency with criminal events i.e. crypto-crime events.

safe investment vehicle may affect buyers' and sellers' preconceptions about the digital money and change their market behaviour, in relation to that sentiment. In this case, the human emotion of fear may have played a role Bitcoin's volatility due to the overly negative discourse of crime being connected to cryptocurrency by the international news press.

## 5.3. The crypto-governance discourse

The LDA output identified topics 3, 5, 6, 9, 11 and 17 relating to a financial governance discourse important in news media (figure 2, LDA topics). Broadly across the entirety of the news corpora in the dataset, the crypto-governance discourse, which as a category can be described as containing news media articles, refers to governing or regulating cryptocurrencies. Publicity through news media reports comprising discussion of governance 'events' appear to have a negative price effect on Bitcoin (see above graphs and further descriptions below).

Topic 5 for instance refers to the Cagayan Economic Zone Authority (CEZA). The CEZA is a freeport area that offers a wide spectrum of business undertakings from beach resorts, world-class golf courses and modern township projects to manufacturing, online gaming and financial technology services for cryptocurrency and Bitcoin companies [48]. Raul Lambino, administrator, and CEO of CEZA, pledged transparency, no corruption, and smooth operations in a bid to attract investment and companies to the area. Asian media outlets such as the Manila Bulletin, Businessworld and the Philippine Star reported on the development of the CEZA zone during 2018 until 20 September 2019, where the various media outlets reported the rise in tax payments to the government-owned corporation, due to development and plans for Chinese businesses to invest $3.9 billion into the CEZA economy. However, in September 2019, the Philippine Star reported that Mike Gerald David, spokesperson and chief fintech and cryptocurrency business officer at CEZA, had stated in a press conference that the agency was suspending the operation of all crypto licensees in Manila by freezing them [49].

The day after publication of the CEZA suspension of crypto licences, the price of Bitcoin fell −12%, which can be seen in the 2019 graph above during September. Since the freezing of licences, the CEZA website claims that CSEZFP will be the first economic zone in Asia to regulate, license and propagate offshore financial technology solutions enterprises and offshore virtual currency exchanges [50]. The 2019 CEZA coverage of the development of the economic zone donning headlines such as 'Crypto Valley of Asia is a haven for foreign investors' [48] would have engendered trust and provided positive sentiment to potential foreign crypto-based investors investing in the area with the promise of state governance, transparency and protection [49]. However, undermining this speculative narrative with one of fear of devaluation and suspicion with the state's freezing of crypto licenses some months later, appeared to correspond with a market downturn in Bitcoin price.

The growth of all crypto activity, however, in Central and Southern Asia[2] from December 2019 to June 2020 rose by an increase in crypto transactions. The value received by Central and Southern Asia[3] rose from 2 billion in December 2019 to over 4 billion in June 2020 [45]. This twofold increase in crypto trading value occurred while investor behaviour responded to a regulatory narrative of crypto licence suspension which had an influential effect on crypto markets, in this case specifically Bitcoin's price. A regional increase in trading at this level could suggest a disproportionate level of media influence on investor behaviour compared with other regions. If, for example, there is an increase in crypto trading in a particular region coinciding with a regional media narrative acting as an amplified social signal affecting regional investor sentiment, this could result in a disproportionate market effect, for example on price formation which appears to be evidenced by Bitcoin's price volatility in the case of Asia's reporting on the CEZA crypto-friendly investment zone.

The market effect may have been regional but in fact could go beyond that region affecting the wider crypto market globally. Just as in the Canadian case with the Quadriga Scandal, the weight of the negative sentiment correlated with a downturn in Bitcoin price which may have been driven by the fact that North America is the third-most active region by cryptocurrency volume moved on-chain [45]. Thus, this may have also been true in the Asian case where negative sentiment again emanated and amplified from regional press coverage, in a geographical area where crypto activity is rising in volume, contributed to falling Bitcoin prices post-publication of each article on the CEZA suspension of crypto licences.

The graph above also shows a governance event that was reported in news media during the 11 July 2019 headlined 'The Future Regulation of Cryptocurrency', which resulted in a subsequential −6% drop

[2]Including Oceania.

[3]Including Oceania.

in the Bitcoin market (as shown in the 2019 graph above). This event referred to the financial action task force issuing guidelines on cryptocurrency which also coincided with Facebook's announcement that it will launch its own brand of cryptocurrency—the Libra.

Of all the governance events that appeared to have a negative effect on the Bitcoin market, the governance event on 15 January 2018 was the most impactful, with a −17% fall in Bitcoin price the following day (as shown in the 2018 graph above). This event related to regulation in South Korea, in a market response to news media headlines such as 'South Korea vows to regulate cryptocurrency'. A week later, more news reports covered the governance further when they released information on the plans to regulate cryptocurrency in South Korea, and on 22 January 2018 the financial authorities reported that the government plans were to require cryptocurrency exchanges to share users' transaction data with banks, in a potential move to impose taxes on the transactions. This resulted in a −6% fall in Bitcoin price (as seen in the 2018 graph shown above), as investors most likely had concerns over privacy and financial ramifications of this news.

Aggressive governance regulation (in the form of a potential ban) was also reported with respect to China's crackdown on cryptocurrency. International news articles covered this governance event happening in China. On 19 November 2018, the Bitcoin price fell −15% (seen in the 2018 graph), in response to published articles on a potential ban on cryptocurrency in China, a key economic powerhouse and cryptocurrency mining hotspot. The United Kingdom regulators had also homed in on attempting to warn of the governance of cryptocurrency earlier in the year during January 2018. While not officially regulating through the policy of a crypto ban, news media (the Independent) recounted the Head of Financial Conduct Authority's warning to Bitcoin investors who do end up buying the cryptocurrency legitimately that they should be prepared to 'lose all their money' due to potential fraud and their respective lack of regulation [51].

Repeated warnings of Bitcoin investment due to lack of regulation were also witnessed in India. For example, news media covered the Reserve Bank of India's warning against investment in cryptocurrencies, which on 4 February 2018 coincided with a −11% drop in the Bitcoin market (as seen in the 2018 graph above). Likewise, news reports covered the Bank of Uganda's warning that investors in the cryptocurrency markets had neither investor protection, nor regulatory purview. This warning coincided with a −6% fall in Bitcoin price on 22 February 2018 (as seen in the 2018 graph above). Further, a few weeks later, on 7 March 2018, a news article in the United Arab Emirates also warned of lack of regulation, which 24 h later corresponded with a −7% Bitcoin price change in the market (as seen in the 2018 graph above). Cryptocurrency governance was also reported on in Thailand's press in addressing the allure of digital asset investment with the discussion of legislation that would regulate digital asset trading and transactions in the Asian country. Subsequently, after this article was published, the Bitcoin price fell by −7% (seen in the 2018 graph).

Governance legislation on cryptocurrency that came into force in Korea appeared to influence the Bitcoin market too. News commentary on the revised bill of the Act on Report and Use of Specific Financial Transaction Information on virtual assets which was approved at a cabinet meeting hit the press in June 2020. The bill outlined the report system for cryptocurrency exchanges and requires virtual asset operators to report their business to the financial authorities. This news corresponded with a −6% Bitcoin price change on 11 June 2020, evidenced in the 2020 graph above.

Additional governance events that appeared to have a negative effect on the Bitcoin market were the news media reports relating to Canada's capital markets watchdogs, putting companies on notice after observing a wave of 'problematic' promotional activities in the cryptocurrency sector. This coincided with a -6% Bitcoin price change on 30 November 2018 (seen in the 2018 graph), with cryptocurrency possibly being viewed by institutional investors as a risky position for company image with the negative press associated with cryptocurrency marketing. In a similar vein to equity research [24], where there is an over-reaction to news that affects investor behaviour, this same sentiment could also apply to the crypto markets, where an over-reaction to media reporting has an impact on investor decisions and their consequential actions to buy, hold or sell their asset in the respective market. The evidence above from the case studies in the sample indicates that crypto-governance events that are reported on in the media lead to a volatile price drop in the Bitcoin market.

## 5.4. The crypto-economy and markets discourse

The LDA output identified topics 0, 2, 5, 6, 9 and 17 relating to the crypto-economy and markets discourse as a salient theme in news media (figure 2, LDA topics). The most significant loss in value for Bitcoin in the period 2018–2020 occurred on 12 March 2020, where the Bitcoin price fell −39% (seen in the 2020 graph). Predictably, this was in line with the stock market in which on the same day,

Wall Street and the FTSE 100 faced the biggest plunge in value since 1987 [52], with the public announcement of the coronavirus pandemic. Aside from this unforeseeable event, the crypto-economy and market discourse had less of an impact on the Bitcoin price than the crypto-crime and crypto-governance discourse. For example, only five published news articles relating to the crypto-economy/ markets corresponded with a high Bitcoin price fall that was more than a 5% drop in price. These five articles were news reports on general market conditions, more financially technical in nature than 'event driven' news. This discourse has been included in this discussion as it arose out of the statistical analysis of the 5% most negative price changes in a two-year period. However, it is important to highlight that it appears to be a less sensitive theme in terms of its associated volatility to the market, due to its lack of prevalence compared to the crypto-crime and crypto-governance discourses, in which the latter featured in a higher frequency of news media articles.

# 6. Conclusion

To conclude, this study built on previous research and gained insight from Mai et al.'s [18] analyses of media and Bitcoin price to create new perspectives taking into account traditional news media and its role in Bitcoin's volatility. This study therefore provided empirical evidence that the key topics associated with cryptocurrency in the international news media within the period 2018–2020 were framed around the following categorical macro discourses: crypto-related crime, financial governance and economy and markets. LDA topic modelling was used as a computational methodology to identify and model data-driven discourses of cryptocurrency in the news media to infer potential 'social signal' or 'sentiment' effects this had on the cryptocurrency markets during the given period. This contributes to the current research and understanding around cryptocurrency [9,29], where past research has revealed a clear connection between the role of the media and the market, i.e. commonly used words and market behaviour [15–17,19].

Building upon previous crypto studies, this analysis provided specific examples of news media presence according to each discourse, namely the crypto-crime, crypto-governance and crypto-economy/markets. Further, it showed possible correlations between discourse theme (or sentiment) and crypto price volatility. The analysis indicated that the crypto-crime, crypto-governance and crypto-economy/markets discourse had a negative effect on Bitcoin prices over a two-year period.

Further, the discussion that followed the analysis explored how each discourse might have differing effects on the crypto markets.

While discourse may have played a role in crypto markets, this study discovered that a potential important factor in the effect of the media on crypto markets may be driven and amplified dependent upon the geographical source of the news. Use of existing data from the 'Geography of Cryptocurrency' report and historical price data complemented the analysis to contextualize discourses and consider the potential weight of sentiment depending on crypto activity in their respective regions. Further research could give more time and consideration to modelling theses crypto discourses to generate statistical trends among the specific discourses identified, to uncover relationships between discourse, source of news and price volatility based upon sentiment. This would also positively add to the budding literature on the role of the media and crypto markets.

Data accessibility. Data and relevant code for this research work are stored in GitHub: https://github.com/kellyann88/Crypto_NLP and have been archived within Zenodo repository: https://doi.org/10.5281/zenodo.6337708.
Authors' contributions. K.C.: conceptualization, data curation, formal analysis, funding acquisition, investigation, methodology, project administration, resources, software, supervision, validation, visualization, writing—original draft and writing—review and editing.
Conflict of interest declaration. I have no competing interests.
Funding. We received no funding for this study.

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
