## [Peer Review File · Royal Society Open Science]

Review History

RSOS-211185.R0 (Original submission)

Review form: Reviewer 1

Is the manuscript scientifically sound in its present form?

No

Are the interpretations and conclusions justified by the results?

No

Is the language acceptable?

Yes

Do you have any ethical concerns with this paper?

No

Have you any concerns about statistical analyses in this paper?

No

Recommendation?

Reject

Comments to the Author(s)

The paper try to analyze the impact that news media have on cryptocurrency prices by leveraging a LDA topic modeling analysis over a set of 4,218 english news articles. The main issue of this work is that the problem is faced in a very superficial way and without providing robust elements to support the theses argued by the author. Here are some of the major problems found:

- A "Related work" section is completely missing from the article. This section should describe the state of the art on this specific research topic, giving insights to which are the most important works, what are the most important finding founds, in which ways your work is different from other works, etc. Without a state-of-the-art review of such type, it is almost impossible to evaluate properly the level of novelty and contributions provided by the work on the reference research field.
- The analysis done on the data is extremely thin and weak and does not support in any way the theses the author want to prove. The fact that some examples of correlation between news and price of a cryptocurrency are shown is not statistically significant (it could simply be due to chance) and it is not clear in how many cases those conditions occur. For example, in the case of topic 1 "The Quadriga Scandal", the author mentions that there are 70 news related to this scandal but reports 3 events that refer to the usual topic (for a total of 9 news) in which the price of Bitcoin seems to fall. What happens in all the other remaining 61 news? What kind of influence do they have on the market?
- One thing that is not considered in this paper and that instead should have a great influence on the news/sentiment/price correlation is how recent a news is, the first time it is released on the market it should have a greater impact (positive or negative) on public opinion while the same news (maybe refreshed with other details) released months later should have a much smaller emotional impact on the market since it is already known to all investors. Based on these considerations, it would also be extremely useful to consider the time factor in the release of news and perhaps verify if and how the impact of the same news changes over time.

In order to greatly improve the quality of the paper, I suggest the author to reorganize/rethink its work by taking inspiration (at least for the provided level of analysis details) from the following relevant works:

-

https://www.researchgate.net/publication/338032363_From_Bitcoin_to_Big_Coin_The_Impacts_of_Social_Media_on_Bitcoin_Performance

- <https://www.tandfonline.com/doi/abs/10.1080/07421222.2018.1440774>

- <https://ieeexplore.ieee.org/abstract/document/9120022>

Review form: Reviewer 2**Is the manuscript scientifically sound in its present form?**

Yes

Are the interpretations and conclusions justified by the results?

Yes

Is the language acceptable?

Yes

Do you have any ethical concerns with this paper?

No

Have you any concerns about statistical analyses in this paper?

No

Recommendation?

Major revision is needed (please make suggestions in comments)

Comments to the Author(s)

I really enjoyed reading this well-referenced and readable paper which explores how media articles can potentially influence cryptocurrency prices through an analysis of thousands of news articles. Specifically, LDA is used to identify 18 topics which correspond to varieties of discourse (e.g. cryptocrime), and then empirical studies are made around related specific news articles and corresponding price movements in cryptocurrencies like Bitcoin and Monero. It *seems* negative news stories tend to impact Bitcoin negatively over the course of several days while there tends to be no such impact on Monero which is a more privacy-focused cryptocurrency.

My main two concerns about the paper are:

1. The topics discovered by the LDA are somewhat "backwards looking" in the sense that they tend to refer to specific news stories (e.g. Quadriga cryptocurrency exchange) rather than general kinds of events; to what extent they could be used to classify future news stories (and thereby future price movements) is therefore questionable.
2. The findings are narrowly focused on specific historical events/topics (predominantly ones driven by negative sentiment), and while there are trends (identified above), it is not clear if these are occurring by chance due to small sample effects or if there are genuine statistical trends at work. Indeed, as acknowledged in Footnote 10 on page 8, there would be great benefit in generalising the analysis. Given the dataset available for this study, I wonder if it is not possible to do this within the scope of this study.

I would recommend that the author address the above two points in a major revision that can then be re-considered.

Typo:

"underly" -> "underlie"

Decision letter (RSOS-211185.R0)

Dear Miss Coulter

The Editors assigned to your paper RSOS-211185 "The Impact of News Media on Cryptocurrency Prices: Modelling Data Driven Discourses in the Crypto-Economy" have made a decision based on their reading of the paper and any comments received from reviewers.

Regrettably, in view of the reports received, the manuscript has been rejected in its current form. However, a new manuscript may be submitted which takes into consideration these comments.

We invite you to respond to the comments supplied below and prepare a resubmission of your manuscript. Below the referees' and Editors' comments (where applicable) we provide additional requirements. We provide guidance below to help you prepare your revision.

Please note that resubmitting your manuscript does not guarantee eventual acceptance, and we do not generally allow multiple rounds of revision and resubmission, so we urge you to make every effort to fully address all of the comments at this stage. If deemed necessary by the Editors, your manuscript will be sent back to one or more of the original reviewers for assessment. If the original reviewers are not available, we may invite new reviewers.

Please resubmit your revised manuscript and required files (see below) no later than 17-Apr-2022. Note: the ScholarOne system will 'lock' if resubmission is attempted on or after this deadline. If you do not think you will be able to meet this deadline, please contact the editorial office immediately.

Please note article processing charges apply to papers accepted for publication in Royal Society Open Science (<https://royalsocietypublishing.org/rsos/charges>). Charges will also apply to papers transferred to the journal from other Royal Society Publishing journals, as well as papers submitted as part of our collaboration with the Royal Society of Chemistry (<https://royalsocietypublishing.org/rsos/chemistry>). Fee waivers are available but must be requested when you submit your manuscript (<https://royalsocietypublishing.org/rsos/waivers>).

Thank you for submitting your manuscript to Royal Society Open Science and we look forward to receiving your resubmission. If you have any questions at all, please do not hesitate to get in touch.

on behalf of Professor Mirella Lapata (Associate Editor) and Marta Kwiatkowska (Subject Editor)
openscience@royalsociety.org

Associate Editor Comments to Author:

The reviewers offer slightly divergent views on the paper - in particular the standard of the literature review conducted - and there are concerns that the treatment of the area is a little lacking in-depth. However, as both reviewers offer ways the author may consider modifying their work to improve it, we'd like to suggest that, while the paper is not ready for further consideration at this stage, it may be with a substantial revision. We would welcome the revision but if you would rather seek publication in an alternative venue, this decision will give you that opportunity. If you opt to revise and resubmit to RSOS, the paper will be returned to the two reviewers here for their view.

Reviewer comments to Author:

Reviewer: 1

Comments to the Author(s)

The paper try to analyze the impact that news media have on cryptocurrency prices by leveraging a LDA topic modeling analysis over a set of 4,218 english news articles. The main

issue of this work is that the problem is faced in a very superficial way and without providing robust elements to support the theses argued by the author. Here are some of the major problems found:

- A "Related work" section is completely missing from the article. This section should describe the state of the art on this specific research topic, giving insights to which are the most important works, what are the most important finding founds, in which ways your work is different from other works, etc. Without a state-of-the-art review of such type, it is almost impossible to evaluate properly the level of novelty and contributions provided by the work on the reference research field.

- The analysis done on the data is extremely thin and weak and does not support in any way the theses the author want to prove. The fact that some examples of correlation between news and price of a cryptocurrency are shown is not statistically significant (it could simply be due to chance) and it is not clear in how many cases those conditions occur. For example, in the case of topic 1 "The Quadriga Scandal", the author mentions that there are 70 news related to this scandal but reports 3 events that refer to the usual topic (for a total of 9 news) in which the price of Bitcoin seems to fall. What happens in all the other remaining 61 news? What kind of influence do they have on the market?

- One thing that is not considered in this paper and that instead should have a great influence on the news/sentiment/price correlation is how recent a news is, the first time it is released on the market it should have a greater impact (positive or negative) on public opinion while the same news (maybe refreshed with other details) released months later should have a much smaller emotional impact on the market since it is already known to all investors. Based on these considerations, it would also be extremely useful to consider the time factor in the release of news and perhaps verify if and how the impact of the same news changes over time.

In order to greatly improve the quality of the paper, I suggest the author to reorganize/rethink its work by taking inspiration (at least for the provided level of analysis details) from the following relevant works:

-

https://www.researchgate.net/publication/338032363_From_Bitcoin_to_Big_Coin_The_Impacts_of_Social_Media_on_Bitcoin_Performance

- <https://www.tandfonline.com/doi/abs/10.1080/07421222.2018.1440774>

- <https://ieeexplore.ieee.org/abstract/document/9120022>

Reviewer: 2

Comments to the Author(s)

I really enjoyed reading this well-referenced and readable paper which explores how media articles can potentially influence cryptocurrency prices through an analysis of thousands of news articles. Specifically, LDA is used to identify 18 topics which correspond to varieties of discourse (e.g. cryptocrime), and then empirical studies are made around related specific news articles and corresponding price movements in cryptocurrencies like Bitcoin and Monero. It *seems* negative news stories tend to impact Bitcoin negatively over the course of several days while there tends to be no such impact on Monero which is a more privacy-focused cryptocurrency.

My main two concerns about the paper are:

1. The topics discovered by the LDA are somewhat "backwards looking" in the sense that they tend to refer to specific news stories (e.g. Quadriga cryptocurrency exchange) rather than general kinds of events; to what extent they could be used to classify future news stories (and thereby future price movements) is therefore questionable.

2. The findings are narrowly focused on specific historical events/topics (predominantly ones driven by negative sentiment), and while there are trends (identified above), it is not clear if these are occurring by chance due to small sample effects or if there are genuine statistical trends at work. Indeed, as acknowledged in Footnote 10 on page 8, there would be great benefit in generalising the analysis. Given the dataset available for this study, I wonder if it is not possible to do this within the scope of this study.

I would recommend that the author address the above two points in a major revision that can then be re-considered.

Typo:

"underly" -> "underlie"

===PREPARING YOUR MANUSCRIPT===

===PREPARING YOUR REVISION IN SCHOLARONE===

Author's Response to Decision Letter for (RSOS-211185.R0)

See Appendix A.

Decision letter (RSOS-220276.R0)

Dear Miss Coulter,

I am pleased to inform you that your manuscript entitled "The Impact of News Media on Cryptocurrency Prices: Modelling Data Driven Discourses in the Crypto-Economy" is now accepted for publication in Royal Society Open Science.

on behalf of Professor Mirella Lapata (Associate Editor) and Marta Kwiatkowska (Subject Editor)
openscience@royalsociety.org

Appendix A

4th March 2022

Dear RSOS Reviewers and Editor,

Many thanks for your helpful contribution to my paper. I have taken your comments on board and addressed them as below.

“Reviewer comments to Author:

Reviewer: 1

Comments to the Author(s)

The paper try to analyze the impact that news media have on cryptocurrency prices by leveraging a LDA topic modeling analysis over a set of 4,218 english news articles. The main issue of this work is that the problem is faced in a very superficial way and without providing robust elements to support the theses argued by the author. Here are some of the major problems found:

Point 1:- A "Related work" section is completely missing from the article. This section should describe the state of the art on this specific research topic, giving insights to which are the most important works, what are the most important finding founds, in which ways your work is different from other works, etc. Without a state-of-the-art review of such type, it is almost impossible to evaluate properly the level of novelty and contributions provided by the work on the reference research field.

Point 2: - The analysis done on the data is extremely thin and weak and does not support in any way the theses the author want to prove. The fact that some examples of correlation between news and price of a cryptocurrency are shown is not statistically significant (it could simply be due to chance) and it is not clear in how many cases those conditions occur. For example, in the case of topic 1 "The Quadriga Scandal", the author mentions that there are 70 news related to this scandal but reports 3 events that refer to the usual topic (for a total of 9 news) in which the price of Bitcoin seems to fall. What happens in all the other remaining 61 news? What kind of influence do they have on the market?

- One thing that is not considered in this paper and that instead should have a great influence on the news/sentiment/price correlation is how recent a news is, the first time it is released on the market it should have a greater impact (positive or negative) on public opinion while the same news (maybe refreshed with other details) released months later should have a much smaller emotional impact on the market since it is already known to all investors. Based on these considerations, it would also be extremely useful to consider the time factor in the release of news and perhaps verify if and how the impact of the same news changes over time.

In order to greatly improve the quality of the paper, I suggest the author to reorganize/rethink its work by taking inspiration (at least for the provided level of analysis details) from the following relevant works:

-
https://www.researchgate.net/publication/338032363_From_Bitcoin_to_Big_Coin_The_Impacts_of_Social_Media_on_Bitcoin_Performance

- <https://www.tandfonline.com/doi/abs/10.1080/07421222.2018.1440774>

- <https://ieeexplore.ieee.org/abstract/document/9120022> “

Author's comments

Point 1: I have not included a 'related work' section as I have followed the RSOS word template which does not include a 'related work' section. I have however addressed the contributions this paper has made in the subsection titled:

'2.3 Contributions of this article with LDA Topic Modelling'

In the above titled section, I describe the state-of-the-art works on this specific research topic, give insight to relevant works, and what are the most important findings. The methodology (LDA topic modelling) is explained as being a key contribution, as well as considering international news articles in the analysis where only social media had been considered in previous research. This research is therefore novel and innovative as I claim in this above section.

Points 2 and 3: In addressing this point, I have focused on sourcing Bitcoin data from messari.io to establish percentage price change in a 24-hour period. Daily returns were calculated over a two-year period. Using excel, the data were ranked from highest percentage change to lowest percentage change. The lowest 5% of these observations constitute the subsample. The news media articles from the Bitcoin price subsample were subsequently categorised into three broad but distinct generalised categories that featured in the topic model: crypto related crime discourse (labelled crypto-crime), financial governance and regulation discourse (labelled crypto-governance), and finally economic discourse (labelled economy).

The charts display the 5% most negative changes in a 2-year period. Each data point represents a news media article published, which is categorised by discourse and is labelled with publication date and bitcoin price change. This 2-year period gives a greater depth and strength to the claim that these generalised events through each discourse in international news media have a negative effect on Bitcoin price.

In terms of how news media repeating stories and their differing effects, this study focused specifically on the initial release of news as a snapshot in time. This study was limited by time and resource and therefore only considers market effects on the initial release of news with respect to the related recent event. However, I do believe this is an interesting area of study for future research given more time and resource. This research however is a good starting point for such future analysis.

The suggested readings: I have incorporated this research into the paper and accounted for its analysis which I believe supports the premise that Bitcoin is sensitive to events and their respective news coverage.

Reviewer: 2

Comments to the Author(s)

I really enjoyed reading this well-referenced and readable paper which explores how media articles can potentially influence cryptocurrency prices through an analysis of thousands of news articles. Specifically, LDA is used to identify 18 topics which correspond to varieties of discourse (e.g., crypto crime), and then empirical studies are made around related specific

news articles and corresponding price movements in cryptocurrencies like Bitcoin and Monero. It *seems* negative news stories tend to impact Bitcoin negatively over the course of several days while there tends to be no such impact on Monero which is a more privacy-focused cryptocurrency.

My main two concerns about the paper are:

Point 1. The topics discovered by the LDA are somewhat "backwards looking" in the sense that they tend to refer to specific news stories (e.g., Quadriga cryptocurrency exchange) rather than general kinds of events; to what extent they could be used to classify future news stories (and thereby future price movements) is therefore questionable.

Point 2. The findings are narrowly focused on specific historical events/topics (predominantly ones driven by negative sentiment), and while there are trends (identified above), it is not clear if these are occurring by chance due to small sample effects or if there are genuine statistical trends at work. Indeed, as acknowledged in Footnote 10 on page 8, there would be great benefit in generalising the analysis. Given the dataset available for this study, I wonder if it is not possible to do this within the scope of this study.

Authors comments

Point 1: While I refer to the Quadriga story as a topical crime event, this is just an illustration of a generalised event related to crime (i.e, crypto crime events, or governance events etc) to qualitatively demonstrate its criminal elements for crypt crime or regulatory events for crypto governance in context. I have generalised these events which are charted into each of three key macro discourses-crypto crime, crypto governance, and crypto economic events). This categorisation could then be used to classify future news stories and related price movements.

A balance of quantitative and qualitative analysis in this paper allows it to benefit from a strong mixed method methodology. The qualitative analysis provides context to the events and allows the reader to appreciate why such events would be classified into each category of discourse, considering their topical characteristics. Meanwhile the quantitative methodology provides a rigour to the analysis through the LDA topic modelling which complements the qualitative methodology into statistically interpreting the importance of topics.

Point 2: I agree most of the historical events/topics were predominately ones driven by negative sentiment. I have therefore in a major revision focused purely on this trend of negative sentiment related to these discourses that appeared out of the data. This allowed for a more thorough analysis of how these generalised event driven discourses had a negative impact on Bitcoin's price over a two-year period.

The analysis has been generalised, as I have described in points 2 and 3 above which address the same comments as reviewer 1.

Typo:

"underly" -> "underlie": The typo 'underly' has been corrected to 'underlie'.

I hope this major revision and suggested changes now satisfy the reviewers, along with my explanation comments. Once again, I thank the reviewers (and editor/s) for their invaluable contribution to improve this paper.

Yours faithfully

The Author